

# Anatomical and physiological responses of *Aechmea blanchetiana* (Bromeliaceae) induced by silicon and sodium chloride stress during *in vitro* culture

Rosiane Cipriano[1,2], João Paulo Rodrigues Martins[3], Lorenzo Toscano Conde[2], Mariela Mattos da Silva[4], Diolina Moura Silva[4], Andreia Barcelos Passos Lima Gontijo[2] and Antelmo Ralph Falqueto[1]

[1] Plant Ecophysiology Laboratory, Federal University of Espírito Santo, São Mateus, Espírito Santo, Brazil
[2] Plant Tissue Culture Laboratory, Federal University of Espírito Santo, São Mateus, Espírito Santo, Brazil
[3] Institute of Dendrology, Polish Academy of Sciences, Kórnik, Wielkopolska, Poland
[4] Center for the Study of Photosynthesis, Federal University of Espírito Santo, Vitória, Espírito Santo, Brazil

## ABSTRACT

Salt stress is one of the most severe abiotic stresses affecting plant growth and development. The application of silicon (Si) is an alternative that can increase the tolerance of plants to various types of biotic and abiotic stresses. The objective was to evaluate salt stress's effect *in vitro* and Si's mitigation potential on *Aechmea blanchetiana* plants. For this purpose, plants already established *in vitro* were transferred to a culture medium with 0 or 14 µM of Si ($CaSiO_3$). After growth for 30 days, a stationary liquid medium containing different concentrations of NaCl (0, 100, 200, or 300 µM) was added to the flasks. Anatomical and physiological analyses were performed after growth for 45 days. The plants cultivated with excess NaCl presented reduced root diameter and effective photochemical quantum yield of photosystem II (PSII) (ΦPSII) and increased non-photochemical dissipation of fluorescence (qN). Plants that grew with the presence of Si also had greater content of photosynthetic pigments and activity of the enzymes of the antioxidant system, as well as higher values of maximum quantum yield of PSII ($F_V/F_M$), photochemical dissipation coefficient of fluorescence (qP) and fresh weight bioaccumulation of roots and shoots. The anatomical, physiological and biochemical responses, and growth induced by Si mitigated the effect of salt stress on the *A. blanchetiana* plants cultivated *in vitro*, which can be partly explained by the tolerance of this species to grow in sandbank (*Restinga*) areas.

Corresponding author
João Paulo Rodrigues Martins,
jprmartinss@yahoo.com.br

## INTRODUCTION

Salinity is responsible for multiple effects that reduce the growth, development, and survival of plants, by means of various mechanisms, including alteration of their hydric relations, deficiency or toxicity of ions, and oxidative stress (*Carillo, 2018*; *Hničková et al., 2019*; *Morton et al., 2019*; *Zhu, Gong & Yin, 2019*; *Chung et al., 2020*).
During prolonged exposure to high salinity, plants suffer ionic stress, mainly due to sodium chloride (NaCl), which negatively affects the synthesis of proteins, enzyme activities, and photosynthesis (*Munns & Tester, 2008*; *Zhu, Gong & Yin, 2019*). Salt stress is accompanied by oxidative stress, leading to the production of reactive oxygen species (ROS). These factors contribute to the deleterious effects of salinity on plants (*Acosta-Motos et al., 2015*; *Zhu, Gong & Yin, 2019*). ROS can alter normal cell metabolism through oxidative damage to the organelles and membranes by lipid peroxidation. Plants' antioxidant systems can be stimulated to combat the oxidative injuries induced by salt stress. These responses include the removal of ROS by enzymes such as ascorbate peroxidase (APX), superoxide dismutase (SOD), and catalase (CAT) (*Zhu, Gong & Yin, 2019*; *Jabeen et al., 2022*).

The physiological mechanisms used by plants to minimize the damages caused by stress and reestablish normal growth include processes such as detection and signaling of stress; regulation of metabolism; reduction of stomatal opening, transpiration, and photosynthesis; inhibition of cell division and expansion; and changes in plants' morphology, phenology, and allocation of resources (*Negrão, Schmöckel & Tester, 2017*; *Morton et al., 2019*). In particular, the regulation of ionic homeostasis involves the sequestration of toxic ions, along with the production and accumulation of organic osmolytes in the cytosol, enabling rapid osmotic adjustment and preventing toxicity (*Nikalje et al., 2017*; *Carillo, 2018*; *Hniličková et al., 2019*; *Larbi et al., 2020*).

Physiological studies of salt stress *in vitro* are considered a feasible alternative to represent adverse conditions of the external environment (*Claeys et al., 2014*). Moreover, this method allows for controlling the stress level and reducing the variability of *in vivo* studies (*Lawlor, 2013*). Studies of salt stress have also been conducted under *in vitro* conditions (*Harter et al., 2014*; *Pandey & Chikara, 2015*; *Cantabella et al., 2017*; *Zushi & Matsuzoe, 2017*; *Rezende et al., 2018*; *Javed & Gurel, 2019*), which have demonstrated the advantages of these techniques for the study of plant physiology.

One alternative to reduce the effects of salt stress on plants is the application of silicon (Si) (*Sahebi, Hanafi & Azizi, 2016*). Si is a beneficial element due to its possibly favorable effects on monocots and eudicots (*Martins et al., 2019*; *Zhu, Gong & Yin, 2019*; *Trejo-Téllez et al., 2020*; *Cipriano et al., 2021b*). Although Si is the majority element in the sand ($SiO_2$) (>90%) (*Costa et al., 2020*), its Si availability for plants is low. Many researchers have reported that Si has attenuating effects on abiotic stresses, such as salinity, drought, and toxicity of heavy metals (*Wu et al., 2015*; *Coskun et al., 2016*; *Manivannan & Ahn, 2017*; *Rios et al., 2017*; *Zhu, Gong & Yin, 2019*; *Cipriano et al., 2021b*).

*In vitro* cultivation allows for studying the physiological functions of Si in plants (*Sivanesan & Park, 2014*; *Rezende et al., 2018*; *Martins et al., 2019*; *Cipriano et al., 2021a*; *Cipriano et al., 2021b*). Using Si in the culture medium of plants grown *in vitro* can increase the growth rate and content of photosynthetic pigments (*Asmar et al., 2015*; *Dias et al., 2017*; *Rezende et al., 2018*; *Martins et al., 2019*; *Cipriano et al., 2021b*). The addition of Si can also favor the increased activity of photosynthesis and the antioxidant system (*Rodrigues et al., 2017*; *Manivannan et al., 2018*; *Ribera-Fonseca et al., 2018*; *Cipriano et al., 2021b*). The positive effects of Si in the culture medium of plants can be related to increased absorption of nutrients and higher photosynthetic activity, besides enhancing the morphogenetic

potential of plants' cells, tissues, and organs (*Sivanesan & Park, 2014*; *Zhu, Gong & Yin, 2019*; *Liu, Soundararajan & Manivannan, 2019*; *Liu et al., 2020*).

Among the techniques for detecting physiological disturbances, pulse-amplitude modulation (PAM) chlorophyll fluorescence is frequently used since it can detect stress by alterations in the performance of photosynthetic apparatus (*Yao et al., 2018*). Besides these, studies of the leaf and root anatomy can be important to evaluate the morphological adjustments of plants in response to stressors (*Paez-Garcia et al., 2015*; *Martins et al., 2019*).

In most studies, only the roots are exposed to salt stress. However, for some plant species, such as bromeliads, the leaves are the primary organ for nutrient uptake. This way, abiotic stress agents, such as salt, can cause different responses compared to those species that face exposure only in the roots. In the present study, we chose the species *Aechmea blanchetiana* (Baker) L.B. Smith (Bromeliaceae). The plants of this bromeliad species grow naturally in sandbank (*Restinga*) areas characterized by high salinity. This species contains a central tank (formed of leaves) that accumulates water, detritus, and salt (sea-salt aerosol generated from ocean–wind). In this context, it is crucial to understand which morphophysiological mechanisms allow mitigating the damages induced by salt stress. It is not yet clear how the co-exposure to Si and NaCl can influence the anatomy, performance of the photosynthetic apparatus, and antioxidant enzymes of plants native to sandbank areas. Therefore, the objective of this study was to evaluate the effect of salt stress *in vitro* induced by NaCl and the mitigation potential of Si in *A. blanchetiana* plants.

## MATERIALS AND METHODS

### *In vitro* culture conditions

Side buds of *A. blanchetiana* with a shoot length of approximately 2.5 cm (previously established *in vitro*) were transferred to glass flasks containing 30 mL MS culture medium (*Murashige & Skoog, 1962*), supplemented with 30 g L$^{-1}$ sucrose and 4 µM 1-naphthaleneacetic acid, and solidified with 7 g L$^{-1}$ agar. The initial treatments consisted of two Si levels (0 or 14 µM CaSiO$_3$) added to the culture medium, and the concentrations were chosen following *Martins et al. (2019)*. After 30 days of *in vitro* culture with both Si levels, the next step was performed. This involved adding 30 mL stationary liquid MS medium (at 25% strength) to the flasks, supplemented with different concentrations of NaCl (0, 100, 200, or 300 µM), forming a solid/liquid medium (two phases) and constituting eight treatments (2 Si ×4 NaCl). The NaCl concentrations were chosen through previous tests, in which plants' highest concentration did not induce death. The treatments with two phases were adapted from the methodology of *Cipriano et al. (2021b)*. The experiment was carried out with five explants per flask, and the treatments involving co-exposure (Si and NaCl) occurred for 45 days (75 total days). The pH of all the media was adjusted to 5.8 before autoclaving at 120 °C during 20 min. The plant material was kept in a growth room with a 16-hour photoperiod under LED lamps (Luminaria LED Slim 36W Bi-Volt 2800 lm) at a temperature of 26 ± 2 °C.

## Analysis of the leaf and root anatomy

After culture for 45 days with Si-NaCl co-exposure, anatomical analyses were performed on the first and second fully expanded leaf and on roots (at 0.5 cm from the plant's base) of six different samples per treatment ($n = 6$). The samples were collected randomly and fixed in an FAA solution (formaldehyde, acetic acid, and 50% ethanol in a proportion of 0.5:0.5:9.0) for 72 h and conserved in 50% ethanol (*Johansen, 1940*). All the microtechnique procedures concerning sectioning, cleaning, and staining of the paradermal and cross-sections were according to *Martins et al. (2018)* and *Martins et al. (2020)*. The sections were then observed under an optical microscope (Leica DM5000 B) coupled with a digital camera (Leica EC3) to capture images. The photomicrographs were analyzed using the UTHSCSA-Imagetool® version 3.0 software, calibrated with a microscopic ruler.

## Analysis of the mineral nutrient levels

The tissue samples were prepared by drying the entire plants in a forced-air oven for 72 h at a temperature between 68 and 72 °C. The analyses were conducted with 1 g dry plant material per sample and three repetitions per treatment ($n = 3$). The samples were ground with a Wiley mill and placed in glass jars. To determine the concentrations of potassium (K), calcium (Ca), magnesium (Mg), sulfur (S), boron (B), zinc (Zn), manganese (Mn), iron (Fe), and sodium (Na), the samples were digested in a nitric-perchloric acid solution in 4:1 proportion (*Sarruge & Haag, 1974*). The minerals were quantified using inductively coupled plasma-optical emission spectrometry (ICP-OES; PerkinElmer model Optima 8300 DV). The nitrogen (N) content was measured by digestion in sulfuric acid according to the Kjeldahl method (*Sarruge & Haag, 1974*).

## Analysis of enzymatic activity

To determine the antioxidant enzyme activities, plants were collected after 45 days of growth. The samples were immediately frozen in liquid nitrogen and stored at $-80$ °C until analysis. The activities of superoxide dismutase (SOD; EC 1.15.1.1), ascorbate peroxidase (APX; EC 1.11.1.11), and catalase (CAT; EC 1.11.1.6) were determined in fully expanded leaves and roots from 5 different samples ($n = 5$). Approximately 0.200 g of fresh-frozen leaf or root samples was ground in a mortar and pestle with liquid nitrogen, potassium phosphate buffer (pH 7.8), EDTA 0.1 mM, ascorbic acid 10 mM, and PVPP 2% w/v. The homogenate was centrifuged at 13,000 g at 4 °C for 10 min. Aliquots of the supernatant were used for the enzymatic assays described below.

SOD activity was determined by forming blue formazan, resulting from nitrotetrazolium blue chloride (NBT) photoreduction following *Giannopolitis & Ries (1977)*. SOD activity was detected after incubation under a 15 W fluorescent light for 10 min at 560 nm and expressed as U min$^{-1}$ mg$^{-1}$ protein. CAT activity was determined according to *Havir & McHale (1987)* by the decay of absorbance at 240 nm, using a 36 mM$^{-1}$ cm$^{-1}$ extinction coefficient and expressed as $\mu$mol H$_2$O$_2$ min$^{-1}$ mg$^{-1}$ protein. APX activity was determined by the initial ascorbate oxidation by H$_2$O$_2$ at 290 nm using a 2.8 mM$^{-1}$ cm$^{-1}$ extinction coefficient and expressed as nmol of ascorbate min$^{-1}$ mg$^{-1}$ protein according to *Nakano & Asada (1981)*. Soluble protein was estimated using Bradford's reagent (B6916; Sigma

Aldrich, Burlington, MA, USA), by the Coomassie blue dye-binding protein assay, with bovine serum albumin as the standard, according to *Bradford (1976)*, to calculate specific enzyme activity.

## Contents of photosynthetic pigments

The contents of photosynthetic pigments were quantified by analyzing eight randomly chosen fragments ($n = 8$) according to the method described by *Martins et al. (2019)*. The absorbance was measured using a Genesys[TM] 10S UV-Vis spectrophotometer (Thermo Fisher Scientific, West Palm Beach, FL, USA), conducted at $\lambda = 470$, 645, and 663 nm for carotenoids (Car), chlorophyll *b* (Chl *b*), and chlorophyll *a* (Chl *a*), respectively.

## Measurement of modulated chlorophyll *a* fluorescence

The analyses of photosynthetic efficiency were carried out between 8:00 and 10:00 a.m. by measuring the modulated chlorophyll *a* fluorescence with a PAM-2500 Walz portable chlorophyll fluorometer. The measurements were carried out on the third leaf from the plant's rosette center of 12 plants per treatment ($n = 12$), according to the method (*Kramer et al., 2004*) and described further in *Martins et al. (2020)*. The following variables were obtained: $F_V/F_M$, ETR, ΦPSII, qN, qP, qL, NPQ, ΦNPQ, and ΦNO.

## Analysis of the growth traits

After 45 days of co-exposure to Si-NaCl, the fresh weight was evaluated of the shoots and roots (g plant$^{-1}$) with five repetitions per treatment ($n = 5$), with each repetition consisting of five plants.

## Statistical analysis

The experimental design was completely randomized in a $4 \times 2$ factorial scheme: 4 NaCl concentrations (0, 100, 200, or 300 μM) and 2 Si concentrations (0 or 14 μM). The data obtained were submitted to analysis of variance (ANOVA), and the means were compared by the Tukey test at 5% probability using the SISVAR 5.4 software (*Ferreira, 2011*).

# RESULTS

## Anatomical analysis

Significant differences were observed in the anatomical traits of the roots. The root diameter was influenced only by the saline concentration, being largest at 100 μM and smallest at 0 μM and 300 μM NaCl (Figs. 1A–1I). The thickness of the cell walls of the exodermis was influenced by both factors evaluated. In the absence of Si, the exodermal cell wall thickness was smaller with all NaCl concentrations compared to the control. In turn, in the presence of Si, the values were similar regardless of the concentration of NaCl applied. However, the cell walls were thinner in relation to those of the roots in the control treatment (Figs. 1A–1H and 1I). The number of metaxylem vessels did not differ among the treatments (5.94 ± 0.55).

In the paradermal leaf sections, the stomatal density of the basal region was influenced only by the NaCl concentration. In this region, there was a decrease in the number of stomata with increasing NaCl concentration (Figs. 2A–2H and 2R). However, the density

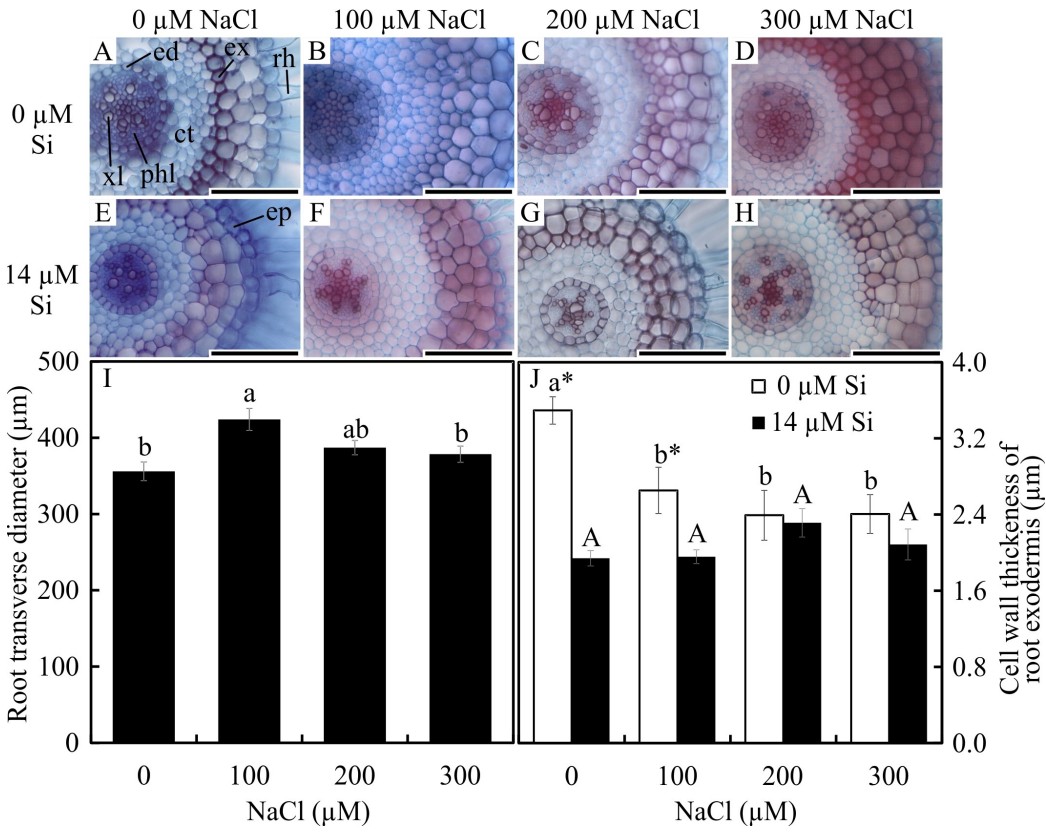

**Figure 1** **Cross-sections (A–H) and anatomical traits (I–J) of roots of *Aechmea blanchetiana* plants in the function of different concentrations of sodium chloride (NaCl) in the absence and presence of silicon (Si) during *in vitro* culture.** Root transverse diameter (I) and cell wall thickness of root exodermis (J) of *Aechmea blanchetiana* in the function of the NaCl concentration ( μM) and in the absence and presence of silicon (Si) during *in vitro* culture. (I) Means (±SE), n = 6, followed by the same letter do not differ according to the Tukey test at 5% significance. (J) Means (±SE), n = 6, followed by the same letter (lowercase for 0 μM Si and uppercase for 14 μM Si), at each NaCl concentration, do not differ according to the Tukey test at 5% significance. For each Si concentration analyzed (0 and 14 μM Si), the means followed by an asterisk are significantly different according to the Tukey test at 5% significance. ct –cortex, ed –endodermis, ep –epidermis, ex –exodermis, rh –root hair, xl –xylem, and phl –phloem. Bars = 100 μm.

and size of the stomata of the middle region of the leaves were influenced only by exposure to Si. When cultivated with 14 μM Si, the plants presented a reduction of 14% in the stomatal density and 3% in the stomatal size (Figs. 2I–2P and 2S–2T).

In the leaf cross-sections, the thickness of the adaxial and abaxial faces of the leaf epidermis (μm) was influenced only by the NaCl concentration, being largest at the concentration of 300 μM NaCl (Figs. 3 and 4A–4B). Plants cultivated in NaCl presence had thicker chlorenchyma, mainly at the concentration of 200 μM NaCl (Figs. 3 and 4C). The area of the sclerenchyma (1007.5 μm² ± 59.86), area of the phloem (587.08 μm² ± 26.01), and diameter of the xylem vessels (9.90 ± 0.36) did not differ significantly among the treatments (Fig. 3).

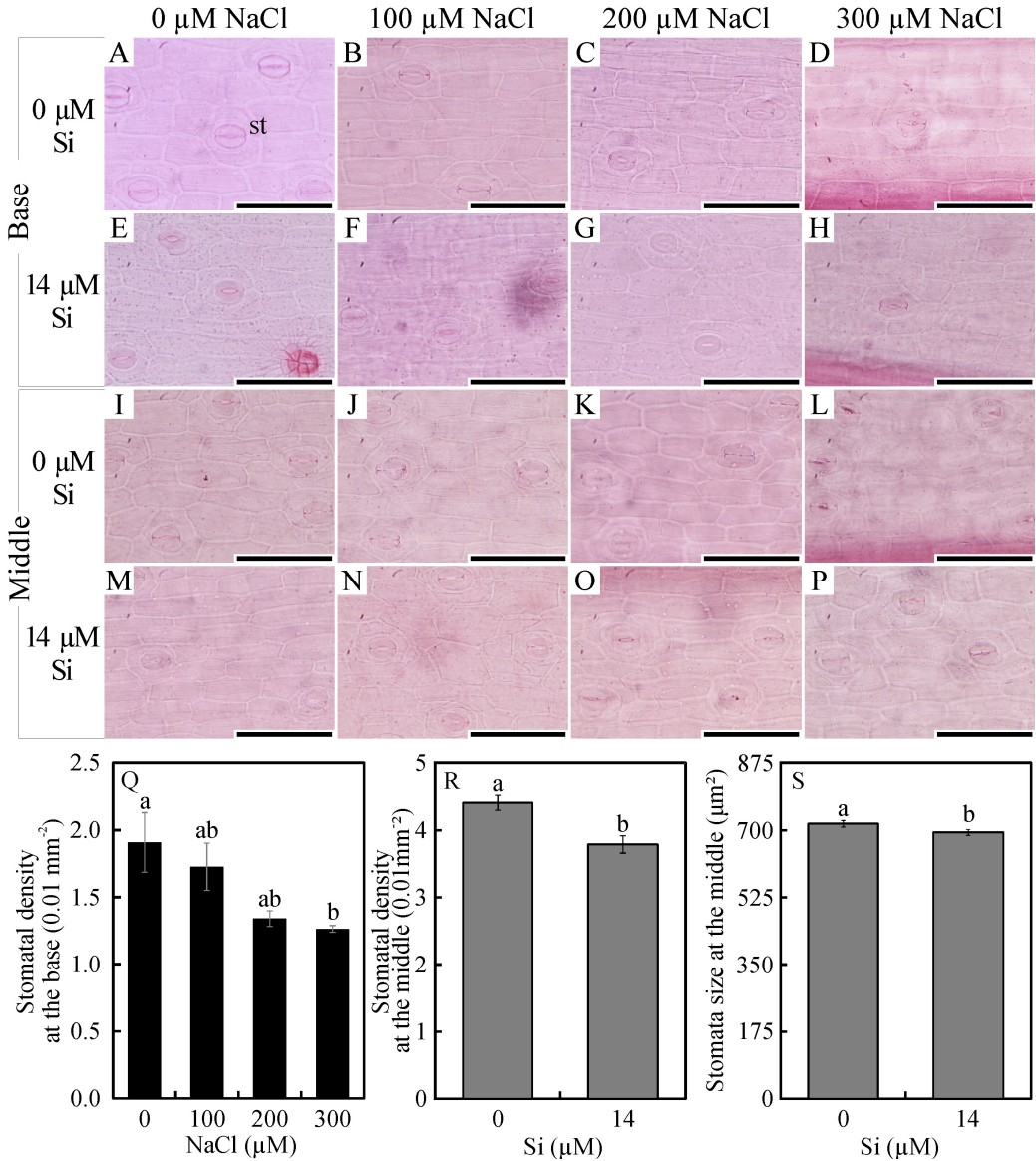

**Figure 2** **Paradermal sections (A–P) and stomatal traits (Q–S) of leaves of *Aechmea blanchetiana* plants in the function of different concentrations of sodium chloride (NaCl) in the absence and presence of silicon (Si) during *in vitro* culture.** Means (±SE), $n = 6$, followed by the same letter, do not differ according to the Tukey test at 5% significance. st –stomata. Bars = 100 μm.

## Contents of nutrients

The contents of Mg, S, Na, and B were influenced by both variation factors, with a significant interaction between them. Plants grown in a medium with Si and NaCl, irrespective of the concentration, had lower content of S. On the other hand, higher NaCl concentration promoted increased Mg, Na, and B contents (Figs. 5A–5D). The contents of Fe, Zn, Mn, and Ca, and the Na/K ratio, were influenced by both variation factors, but Si and NaCl acted independently. Lower contents of Fe, Zn, Mn, and Ca were observed in NaCl presence.

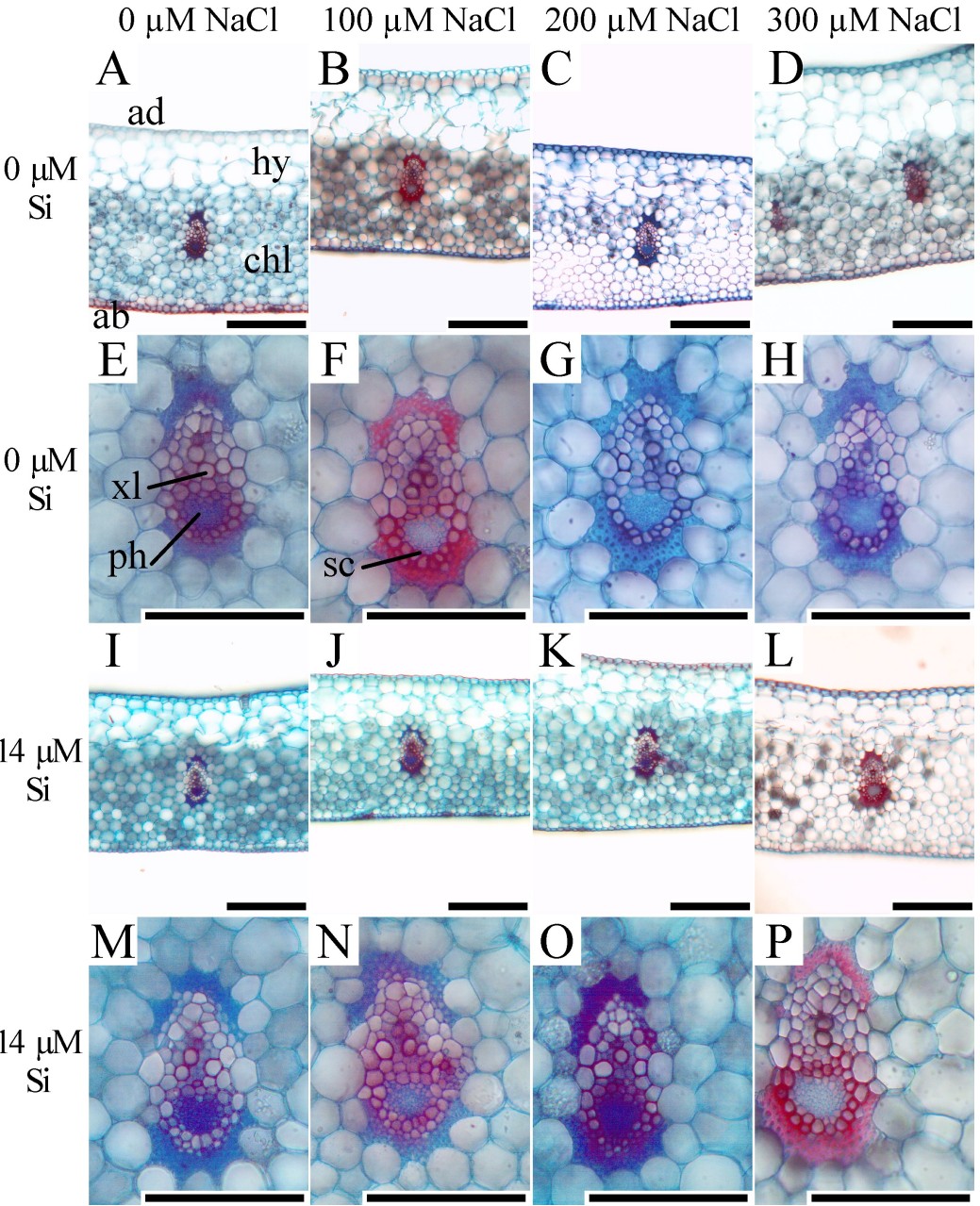

**Figure 3** Cross-sections of leaves of *Aechmea blanchetiana* plants in the function of different concentrations of sodium chloride (NaCl) in the absence and presence of silicon (Si) during *in vitro* culture. ad—adaxial epidermis, ab–abaxial epidermis, chl—chlorenchyma, hy—hydrenchyma, ph—phloem, sc—sclerenchyma, xl—xylem. Bars = 100 μm.

A higher Na/K ratio was associated with greater salt concentration. The content of K did not differ significantly between the different concentrations of NaCl (2.027 ± 0.098), nor did the content of N (1.75 ± 0.073). The presence of Si increased the contents of Zn, Mn,

Ca, and N and the Na/ K ratio. Finally, the content of Fe (179.97 ± 8.77) did not differ in function of the Si concentration (Figs. 6A–6B).

## Antioxidant enzyme activity

The activity of SOD and CAT, both shoots and roots, was influenced by both variation factors. The activity of SOD was higher in the plants cultured with Si and increased in the function of rising NaCl concentration. The greatest activity of SOD occurred in the presence of 300 μM NaCl, both in the leaves and roots (Figs. 7A–7B). The activity of CAT was greatest in the plants grown with Si and increased with rising concentrations of NaCl (200 and 300 μM NaCl) (Figs. 7C–7D). Both factors influenced the activity of APX, but they acted separately. The activity of APX was highest with greater concentrations of NaCl in the plants cultivated in a medium supplemented with Si, both in the leaves and roots (Figs. 7E–7F).

## Contents of photosynthetic pigments

Only the treatment with Si influenced the contents of photosynthetic pigments. Plants cultivated with Si had higher contents of Chl *a*, Car, and Chl total, but there was no alteration in Chl *b* and Chl *a/b* in the leaves of *A. blanchetiana* plants cultured *in vitro* (Fig. 8).

## Analysis of modulated chlorophyll *a* fluorescence

The variables ΦPSII and ETR were influenced only by NaCl, with lower values associated with rising NaCl concentration (Figs. 9A–9B). In turn, NPQ and $F_V/F_M$ were influenced by Si, presenting lower NPQ and higher $F_V/F_M$ in the plants grown in a medium supplemented with Si (Figs. 9C–9D). qP, qL, qN, ΦNPQ, and ΦNO were all influenced by both variation factors. No significant differences were observed among the plants cultivated without Si for qP and qL. However, at the highest concentration of NaCl, there were increases in qN and ΦNPQ. The highest values of ΦNO were obtained in the control plants and those grown with 100 μM NaCl and the lowest at 200 μM NaCl. Among the plants cultivated in the presence of Si, the lowest values of qP and qL were observed in those grown with 200 μM NaCl, as was the case for qN. However, the greatest values of qN were obtained in the plants cultivated in a medium containing 300 μM NaCl. The absence of NaCl was associated with the highest values of qP and qL. No changes in ΦNO and ΦNPQ were observed between treatments (Fig. 10).

## Analysis of growth

The fresh weights of the roots and shoots were influenced by both variation factors. Among the plants grown without Si, the shoot and root weights declined with increasing NaCl concentration. However, among the plants cultivated in a medium with Si, the shoot's fresh weight increased in the presence of 200 μM NaCl, while the root's fresh weight increased in the plants receiving 100 μM NaCl. Overall, the fresh weights of the shoots and roots were greater in the plants cultivated in Si and higher concentrations of NaCl than in those grown without Si (Fig. 11).

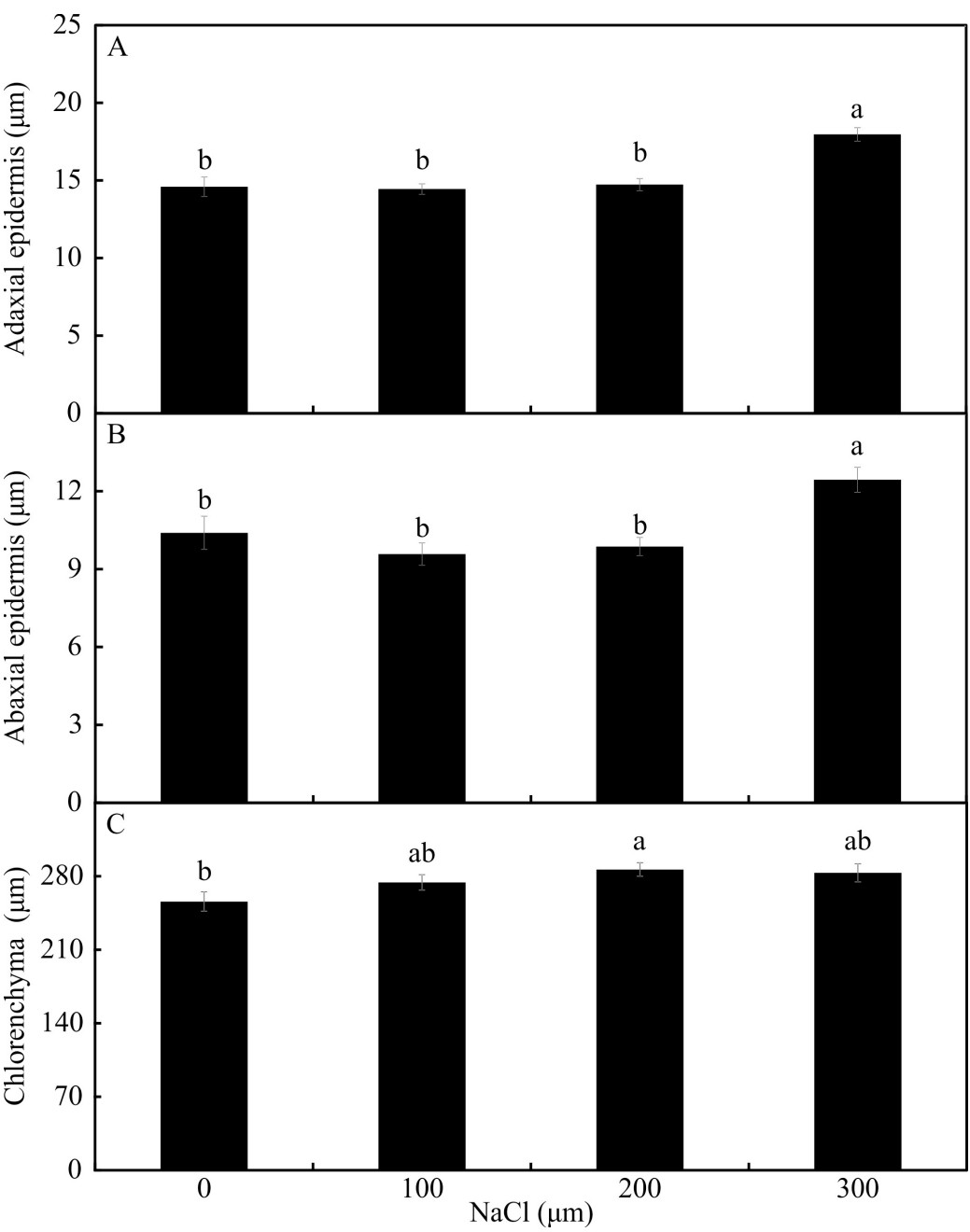

**Figure 4** The thickness of the adaxial and abaxial faces of the epidermis (µM) (A –B) and the chlorenchyma (C) of leaves of *Aechmea blanchetiana* in the function of the concentrations of NaCl (0, 100, 200, 300 µM). Means (±SE), $n = 6$, followed by the same letter, do not differ according to the Tukey test at 5% significance.

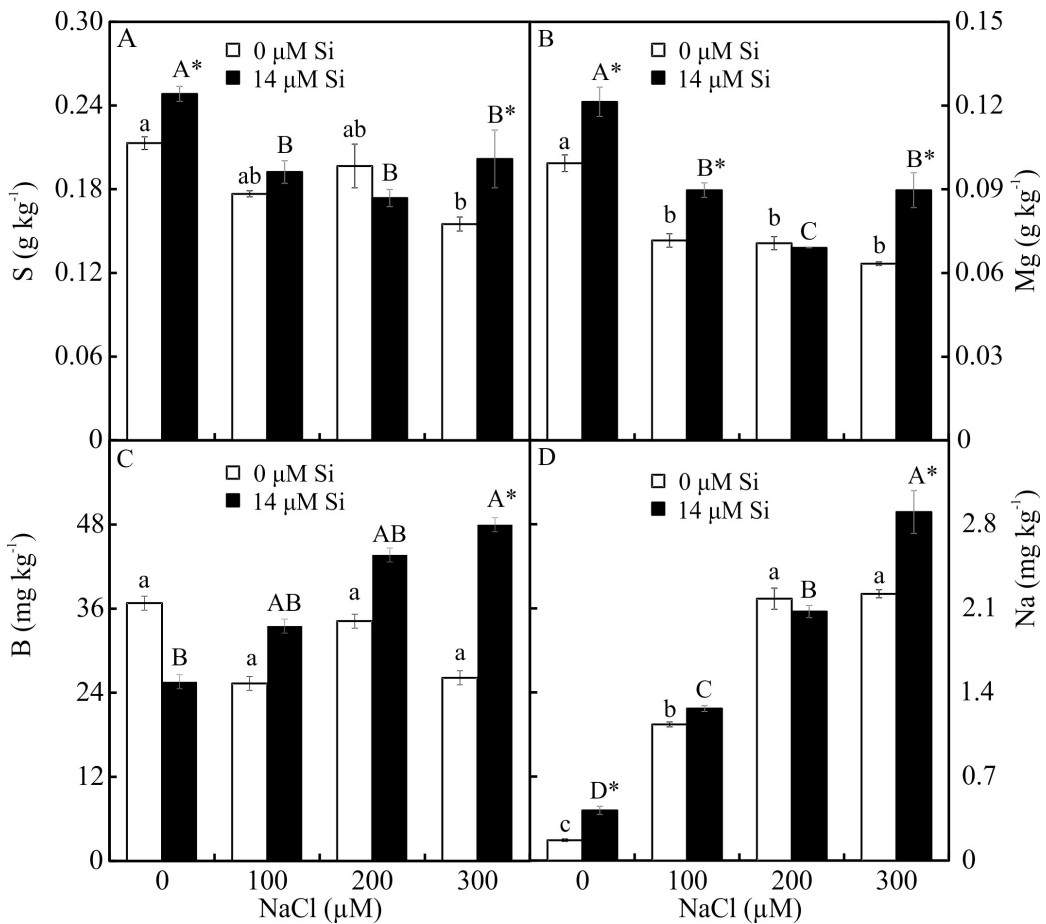

**Figure 5** (A–D) Contents of macronutrients and micronutrients in *Aechmea blanchetiana* plants in the function of NaCl concentrations (0, 100, 200, 300 µM) combined with 0 or 14 µM Si. For each nutrient, the means (±SE), $n = 3$, followed by the same letter (lowercase for 0 µM Si and uppercase for 14 µM Si), at each NaCl concentration, do not differ according to the Tukey test at 5% significance. For each Si concentration analyzed (0 and 14 µM Si), the means followed by an asterisk are significantly different according to the Tukey test at 5% significance. S = sulfur, Mg = magnesium, B = boron, Na = sodium.

## DISCUSSION

The *A. blanchetiana* plants cultivated under the *in vitro* conditions imposed showed different anatomical and physiological responses due to the presence or absence of Si and the variation in concentrations of NaCl. The morphophysiological responses induced by Si had an attenuating effect on salt stress, through anatomical alterations, increased content of photosynthetic pigments, and greater activity of the enzymes of the antioxidant system, besides their contribution to enhance the performance of the photosynthetic apparatus.

The root and leaf anatomy of the plants was in accordance with the previous description by *Martins et al. (2018)*. The reduction of the diameter of the root cross-sections under salt stress conditions found in this study might have resulted from reductions in the size and number of cells, especially in the internal cortex. The alterations of the cell size can

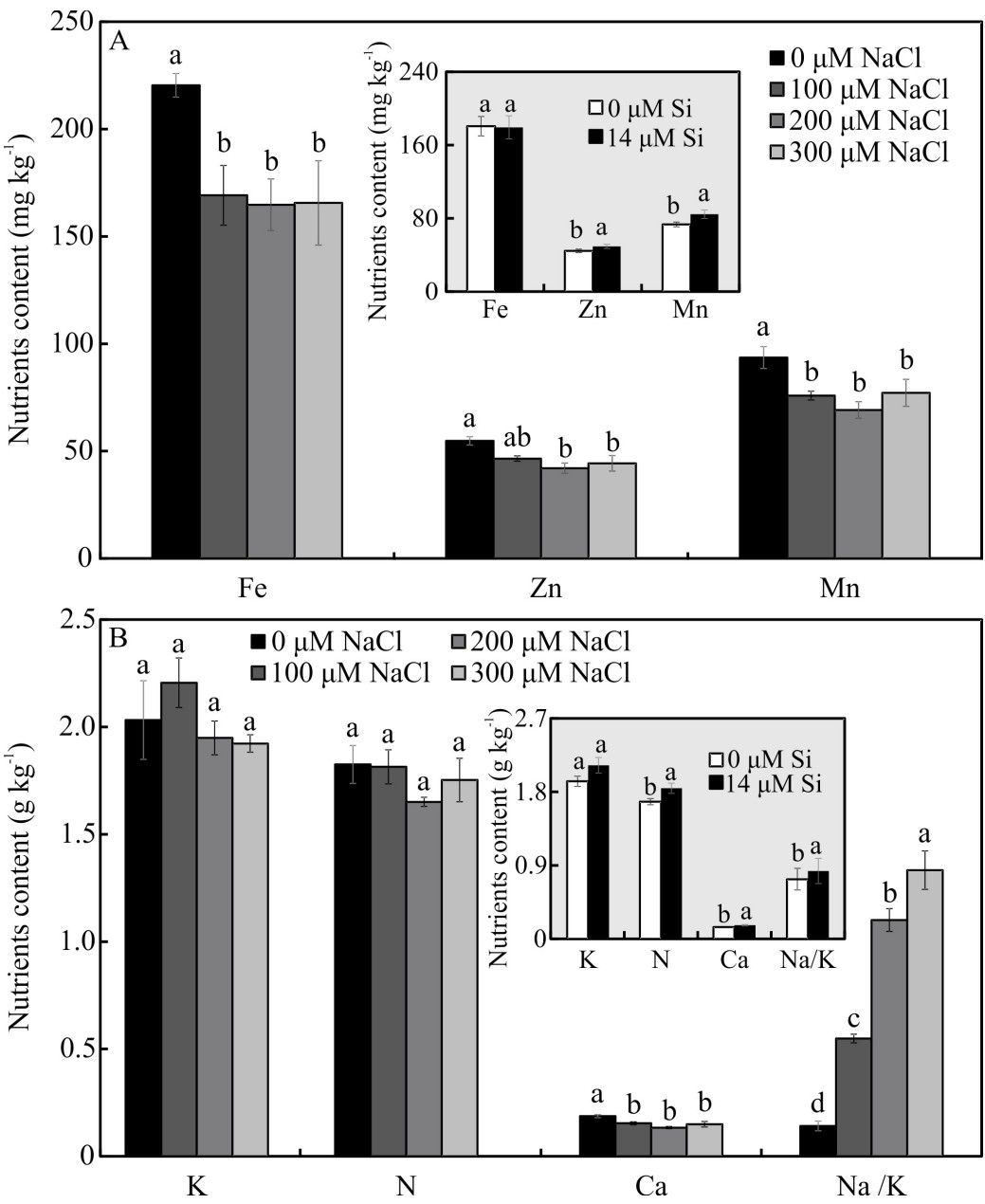

**Figure 6** (A–B) Contents of nutrients in *Aechmea blanchetiana* plants in the function of the concentrations of NaCl (0, 100, 200, 300 μM) or concentration of Si (0 or 14 μM Si). For each content of nutrients, the means (±SE), $n = 3$, followed by the same letter do not differ according to the Tukey test at 5% significance. Fe = iron, Zn = zinc, Mn = manganese, Ca = calcium, N = nitrogen, K = potassium, Na = sodium.

be related to resistance to salt stress since smaller cells can indicate an essential response to increase the water potential, possibly contributing to more effective maintenance of turgor under water deficit (*Munns & Tester, 2008*; *Terletskaya et al., 2019*). Reduced root

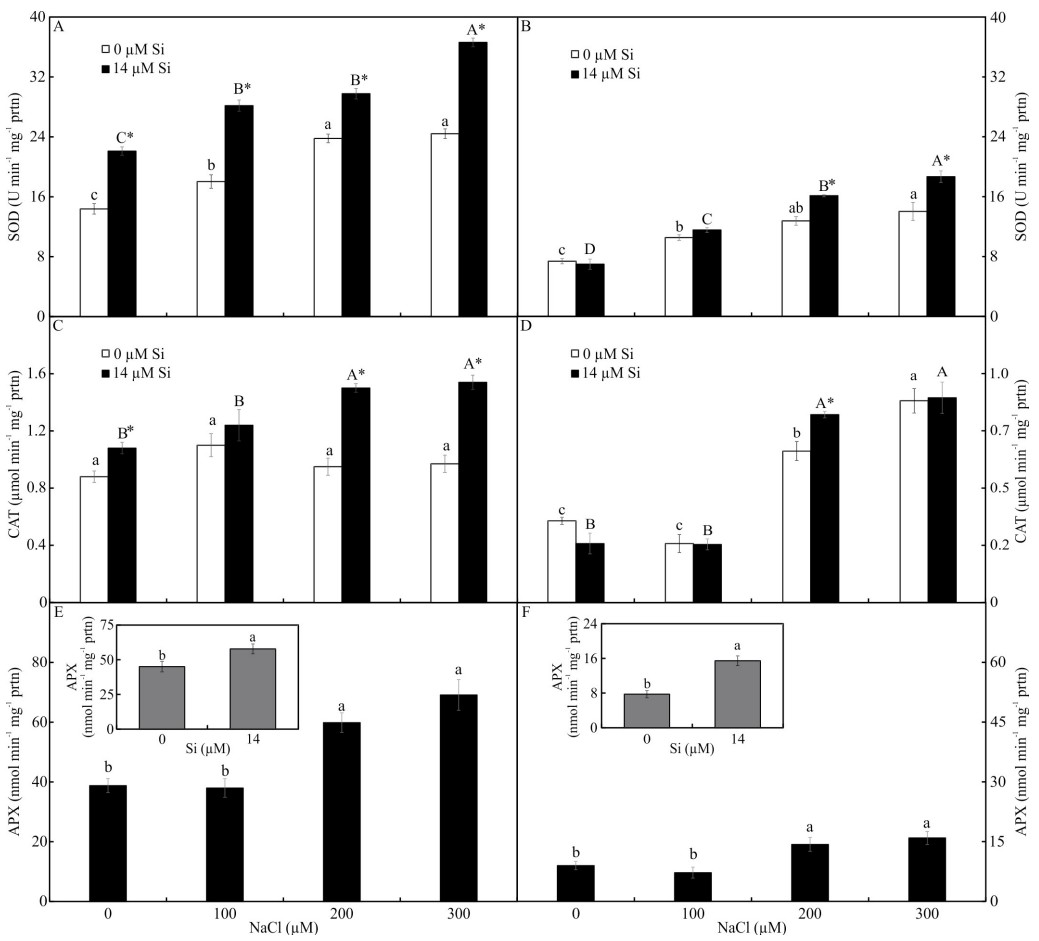

**Figure 7 Antioxidant enzyme activity in the leaves (A-C-E) and roots (B-D-F) of *Aechmea blanchetiana* plants cultivated *in vitro* in the function of NaCl and Si.** Means (±SE), $n = 5$, followed by the same letter (lowercase for 0 μM Si and uppercase for 14 μM Si), at each NaCl concentration, do not differ according to the Tukey test at 5% significance. For each Si concentration analyzed (0 and 14 μM Si), the means followed by an asterisk are significantly different according to the Tukey test at 5% significance (A–D). Means (±SE), $n = 5$ followed by the same letter do not differ according to the Tukey test at 5% significance (E–F).

diameters can be a sign of adaptation to the high pressure of the water column on the conductor system (*Rewald et al., 2013*; *Terletskaya et al., 2019*).

The induction of a thinner exodermis observed in this study in response to excess NaCl in the shoots may have been the key to the NaCl tolerance. It may induce a greater flow of nutrients from the culture medium to the shoots, improving the nutritional balance. This thickening occurs naturally by the deposition of lignin and/or suberin, and the degree of thickening can moderate the uptake and translocation of mineral nutrients to the entire plant (*Martins et al., 2019*). Thus, the reduction in the thickness of the exodermis cell walls caused by Si shows that this element acted positively, facilitating the uptake of nutrients from the culture medium.

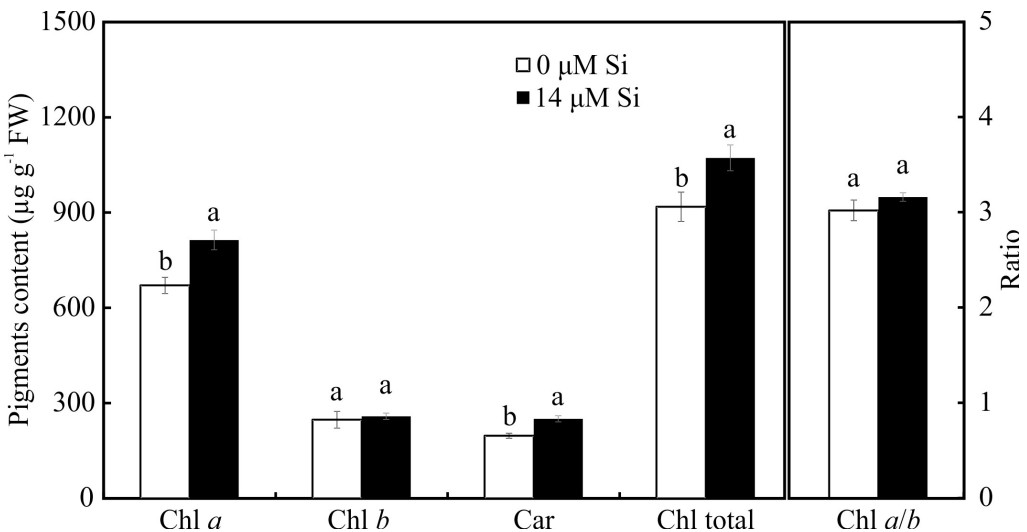

**Figure 8 Contents of the photosynthetic pigments in *Aechmea blanchetiana* plants in the function of the presence or absence of Si (0 or 14 μM Si).** Means (±SE), $n = 8$, followed by the same letter in each photosynthetic pigment, do not differ according to the Tukey test at 5% significance.

In the leaves, the direct exposure to NaCl at the leaf base reduced the stomatal density. Besides this, the epidermis was thicker in the plants exposed to salt. These responses together suggest a morphological adjustment to control the entry of NaCl through the symplastic and transcellular veins (*Morton et al., 2019*). Considering that plants can also uptake nutrients through the leaves, an increase in the thickness of the epidermis can act as a mechanism to control the absorption of excessive NaCl (*Mahmood et al., 2019*). It has been suggested that the movement of nutrients to the interior of plants can involve diffusion through the cuticle and absorbed by leaf cells. Absorption through the stomatal pore can also occur since the stomata act as potential pathways for the movement of nutrients applied to the leaves (*Li et al., 2019*).

In the middle region of the leaves, the stomatal density was greater than in the base region, as previously observed by *Santos et al. (2020)*. However, a comparison of the treatments revealed that Si could influence the morphology of the stomata of other leaf regions. The morphophysiological modulations in the middle leaf region in plants grown with Si, such as smaller stomatal density and size, might have occurred to reduce the osmotic stress (*Mahmoudi et al., 2020*; *Morton et al., 2019*). This reduction resulting from the action of Si might be a mechanism to maintain the prompt functioning of the stomata for osmotic control. The size of the stomata is related to their functionality because smaller guard cells respond (open/close) faster than larger ones, and consequently maintain the stomatal conductance (*Rouphael et al., 2017*). Another alteration observed in this study was an increase in the thickness of the chlorenchyma, apparently related to a tradeoff mechanism in which the smaller leaf area is offset by the greater thickness of this tissue (*Pereira et al., 2016*). This capacity for protecting the photosynthetic tissues permits the maintenance of the plant's biomass production (*Pereira et al., 2016*; *Ribeiro et al., 2019*).

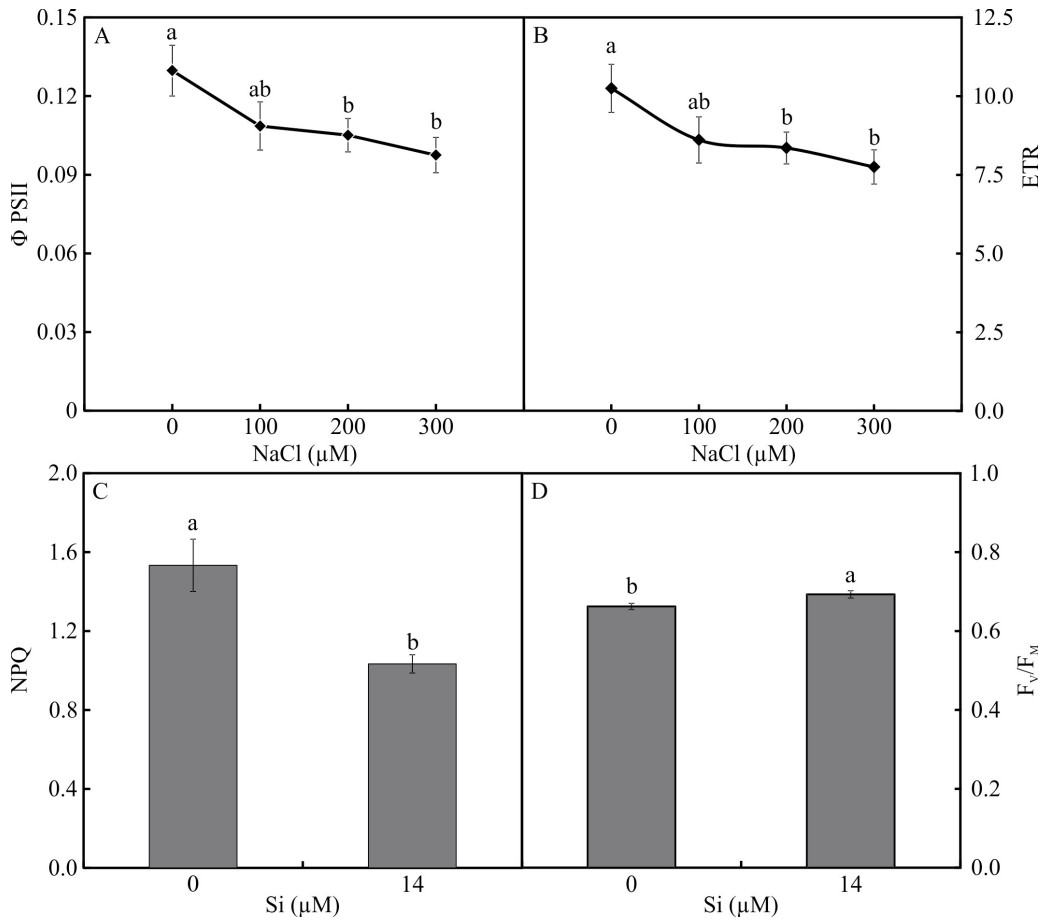

**Figure 9** **ΦPSII (A) and ETR (B) in the function of the concentrations of NaCl (0, 100, 200, 300 μM). NPQ (C) and $F_V/F_M$ (D) in *Aechmea blanchetiana* plants in the function of Si (0 or 14 μM Si).** Means (±SE), $n = 12$, followed by the same letter for each parameter, do not differ according to the Tukey test at 5% significance.

The excess of NaCl altered the content of mineral nutrients in *A. blanchetiana*, reducing the contents of the macronutrients S and Ca and the micronutrients Fe, Zn, and Mn. The excessive accumulation of $Na^+$ competitively inhibits the absorption of other cations, including $K^+$, $Ca^{2+}$, and $Fe^{2+}$, leading to an imbalance in cell homeostasis, oxidative stress, and interference in the functions of $Ca^{2+}$ and $K^+$ (*Kim et al., 2021*). We suggest that reducing the contents of S, Ca, Fe, Zn, and Mn reduced the stress tolerance of the plants, generating oxidative stress and affecting the performance of the photosynthetic apparatus. Limited availability of Ca can reduce the tolerance of plants to salt stress since this is involved in the gene induction of tolerance to salt stress and regulation of the antioxidant defense (*Liu, Soundararajan & Manivannan, 2019*). K plays a fundamental role in synthesizing proteins, photosynthesis, and the activity of glycolytic enzymes in plants (*Liu, Soundararajan & Manivannan, 2019*).

The modulations of the contents of mineral nutrients in *A. blanchetiana* promoted by Si contributed to improve the nutritional balance and mitigated the damages caused by

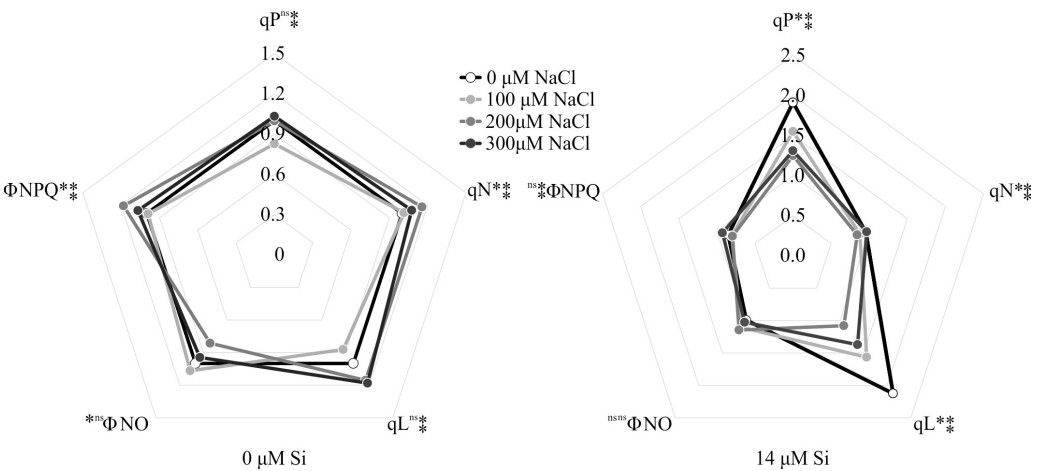

**Figure 10 Modulated fluorescence parameters of *Aechmea blanchetiana* plants in the function of the concentrations of NaCl (0, 100, 200, 300 μM) combined with 0 μM Si or 14 μM Si.** For each parameter, means ($n = 12$) followed by an asterisk (*) denote significant differences between the concentrations of NaCl at each level of Si, while two asterisks (⁂) denote significant differences between the presence and absence of Si according to the Tukey test at 5% probability. ns = no significant.

the toxicity of NaCl in the leaf cells. The increase promoted by Si in the contents of the nutrients Ca, B, Zn, Mn, N, and Mg was probably due to the thinner exodermis in the roots, modulated by Si, which allowed greater absorption of these nutrients. Higher B content may also increase the antioxidant system's defense and diminishes oxidative stress (*Rahman et al., 2021*). These responses resulted in a better nutritional balance contributing to an increase in the content of photosynthetic pigments and the activity of the enzymes of the antioxidant system (SOD, APX, and CAT). This promoted the protection of the plants' tissues against oxidative damage to the membrane under salt stress, thus alleviating the toxicity of salt and increasing the growth of *A. blanchetiana* plants. The increase in the activity of antioxidant enzymes is also responsible for reducing oxidative stress and eliminating the ROS produced during salt stress (*Tewari, Kumar & Sharma, 2019*; *Zhang et al., 2019*; *Chung et al., 2020*; *Kim et al., 2021*). These nutrients are structural components of the chlorophyll molecule and play a role in forming the amino acids necessary for the processes of the antioxidant defense system, acting as enzymatic cofactors, for example (*Rahman et al., 2016*; *Tewari, Kumar & Sharma, 2019*; *Santos et al., 2021*). Besides this, the greater activity of the antioxidant system enzymes leads to lower degradation of chlorophyll (*Gong et al., 2018*). Alterations in the antioxidant system enzymes evidenced the physiological stress caused by NaCl exposure. In this study, the activity of the antioxidant enzymes was greater in the leaves than in the roots of the plants. This result indicates that the direct exposure to NaCl on the leaves had an impact, generating oxidative stress. However, the higher activity of enzymes of plants cultivated in a medium supplemented with Si showed the benefits of this element.

Even though the presence of Na caused stress, as indicated by the biochemical alterations described, this element also appears to play a fundamental role in the metabolism of *A.*

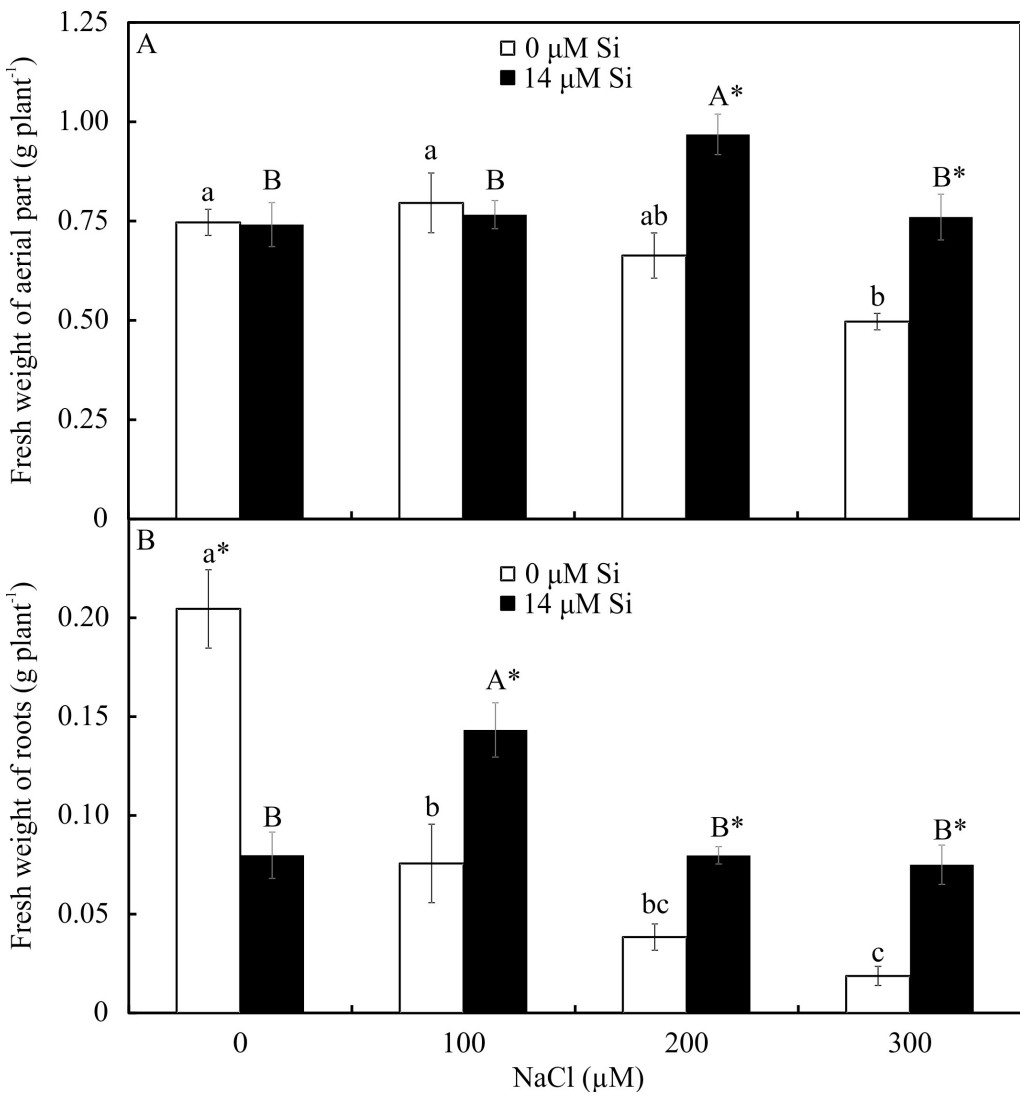

**Figure 11  Shoot (A) and root (B) fresh weights of *Aechmea blanchetiana* plants in the function of the concentrations of NaCl combined with 0 or 14 μM Si.** Means (±SE), $n = 5$, followed by the same letters (lowercase for 0 μM Si and uppercase for 14 μM Si) at each NaCl concentration, do not differ according to the Tukey test at 5% significance. For each Si concentration analyzed (0 and 14 μM Si), the means followed by an asterisk (*) are significantly different according to the Tukey test at 5% significance.

*blanchetiana* plants, and its absorption in minimum quantities seems to have occurred. Plants grown in the 0 μM NaCl + 14 μM Si condition had greater content of Na than in the control plants (Na added only in the form of Na-EDTA in the MS medium). Other studies have shown that *A. blanchetiana* has crassulacean acid metabolism (CAM) for carbon fixation under adverse conditions (*Chaves, Leal & Lemos-Filho, 2015*; *Krause et al., 2016*). We suggest that even in *in vitro* conditions, *A. blanchetiana* plants can have some CAM behavior level, such as reducing leaf area and making the leaves more compact. Plants that use CAM metabolism can require sodium ions ($Na^+$) (*Scholl et al., 2020*). In

this species, Na$^+$ seems to be fundamental for the regeneration of phosphoenolpyruvate, the substrate for initial carboxylation in plants with C$_4$ and CAM metabolism (*Scholl et al., 2020*). CAM metabolism is a mechanism that protects against increased salinity, but the most critical tolerance mechanism can be the accumulation of ions in leaf vacuoles for osmotic adjustment (*Montero et al., 2018*). In halophytes, the accumulation of Na and its compartmentalization in vacuoles modulate the osmotic potential and help guarantee water absorption under salt stress conditions (*Zeng et al., 2015*). The Na ions stimulate growth by promoting cell expansion and partially substitute K ions as an osmotically active solute (*Hussain et al., 2010*). This modulation of the content of Na$^+$ can partly explain the salt tolerance and, thus, the existence of *A. blanchetiana* in the sandbank (Restinga) region studied here. Furthermore, this can also explain the increase in the Na/K ratio with higher salt concentration and the presence of Si observed in this study, which has been confirmed to be one of the main determinants of resistance to salts (*Liu et al., 2020*). Despite this increase in the Na/ K ratio, Si was responsible for modulating the competitive absorption between Na and K and maintaining the balance in the intercellular distribution of K in the *A. blanchetiana* plants since the content of K was not different among the treatments.

The morphophysiological modulations promoted by Si, such as the greater activity of the enzymes SOD, APX, and CAT, reduced the stress on the photosynthetic apparatus, as demonstrated by the analysis of the chlorophyll *a* fluorescence. The plants grown in the medium supplemented with Si had the highest values of qP and qL, implying a more remarkable ability for photochemical conversion and transfer of electrons from PSII (*Wang et al., 2018*). This suggests that even though the plants grown with high NaCl suffered photodamage, the Si was able to ameliorate this damage by maintaining a proper balance of nutrients, as well as enhancing the activity of the antioxidant system, impeding oxidative damage to the photosystems (*Liu et al., 2020*). The Si also contributed to maintain the electron transport, as evidenced by the higher $F_V/F_M$ ratio, indicating greater potential photochemical activity of PSII (*Lotfi et al., 2018*). Factors for the photosynthetic apparatus's functioning were also evidenced by the lower values of the parameters of non-photochemical quenching, such as qN, ΦNPQ, and NPQ, compared with the plants grown without Si. These responses helped reduce the damage to the plants caused by the stress, which in turn helped maintain the plants' growth since they had greater fresh weight when cultivated with higher concentrations of NaCl. The excess of NaCl in plants grown without Si caused increases of qN, ΦNPQ, and ΦNO, leading to over-reduction of the photosynthetic electron transport chain, excess excitation energy, and consequently, reduction of the photochemical step and biochemical processes. Furthermore, the increase of ΦNO indicates that this energy loss did not involve the action of trans-thylakoid ΔpH and zeaxanthin, meaning the excess flow of energy was out of control (*Yao et al., 2018*; *Wang et al., 2018*).

The increased stress level caused by NaCl affected the functioning of the photosynthetic apparatus by reducing the values of ΦPSII and ETR. The decrease might have partly inhibited the transport of electrons and effective photochemical activity of PSII and increased the formation of ROS since the activity of the antioxidant system was affected. This reduction indicates a smaller density of the flow of photons absorbed by PSII

(*Wang et al., 2018*). These responses induced by excess NaCl in the absence of Si caused a reduction in the plants' growth. The decline of the fresh and dry weights of the leaves and roots, and thus the reduction in growth, are symptoms commonly observed in plants under salt stress (*Dias et al., 2017*). This result can be attributed to the osmotic effect of the salt solution beyond the roots, as well as an imbalance in the absorption and assimilation of nutrients (*Dias et al., 2017*; *Rezende et al., 2018*).

## CONCLUSION

In *in vitro* conditions, NaCl acted to stunt the growth of the *A. blanchetiana* plants since it affected the plants' anatomy, uptake of nutrients, and physiology. These plants presented tolerance responses by implementing various mechanisms to deal with salt stress, such as thinner walls of the exodermis, reduced stomatal density, and increased non-photochemical dissipation of fluorescence. The application of Si reduced the damages generated by stress through modulation of the root anatomy, enabling greater uptake of nutrients essential for the antioxidant system's activity. The greater enzymatic activity reduced oxidative stress and enabled alterations in the functioning of the photosynthetic apparatus. These modulations contributed to minimizing the damage to the plants caused by the stress, as proved by the chlorophyll *a* fluorescence.

**Abbreviations**

| | |
|---|---|
| $\Phi$PSII = Y(II) = $\Phi$(II) | Effective photochemical quantum yield of PSII |
| ETR | Rate of linear electron flow |
| $F_V/F_M$ | maximum quantum yield of PSII |
| NPQ | non-photochemical fluorescence dissipation |
| PSII | photosystem II |
| qP | photochemical quenching |
| qL | photochemical fluorescence quenching assuming interconnected PSII antennae |
| qN | non-photochemical quenching |
| $\Phi$NPQ | quantum yield induced light (NpH and zeaxanthin-dependent) from non-photochemical fluorescence dissipation |
| $\Phi$NO | quantum yield of non-regulated energy dissipation |

## ACKNOWLEDGEMENTS

The authors acknowledge Luiz Carlos de Almeida Rodrigues for technical assistance in making the figures.

### Funding

The authors received no funding for this work.

## Competing Interests

The authors declare there are no competing interests.

## Author Contributions

- Rosiane Cipriano conceived and designed the experiments, performed the experiments, analyzed the data, prepared figures and/or tables, authored or reviewed drafts of the article, and approved the final draft.
- João Paulo Rodrigues Martins conceived and designed the experiments, performed the experiments, analyzed the data, prepared figures and/or tables, authored or reviewed drafts of the article, and approved the final draft.
- Lorenzo Toscano Conde performed the experiments, authored or reviewed drafts of the article, and approved the final draft.
- Mariela Mattos da Silva analyzed the data, prepared figures and/or tables, authored or reviewed drafts of the article, and approved the final draft.
- Diolina Moura Silva analyzed the data, prepared figures and/or tables, authored or reviewed drafts of the article, and approved the final draft.
- Andreia Barcelos Passos Lima Gontijo conceived and designed the experiments, authored or reviewed drafts of the article, and approved the final draft.
- Antelmo Ralph Falqueto conceived and designed the experiments, authored or reviewed drafts of the article, and approved the final draft.

## Data Availability

The raw data is available in the Supplemental Files.

## Supplemental Information

Supplemental information for this article can be found online at http://dx.doi.org/10.7717/peerj.14624#supplemental-information.

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
