# Peer review of "Anatomical and physiological responses of Aechmea blanchetiana (Bromeliaceae) induced by silicon and sodium chloride stress during in vitro culture"

_PeerJ, doi:10.7717/peerj.14624_

## Round 0.1 · original submission · Major Revisions

Dear Dr. Martins,
Thank you for your submission to PeerJ.
It is my opinion as the Academic Editor for your article - Anatomical and physiological responses Aechmea blanchetiana (Bromeliaceae) induced by silicon and sodium chloride during in vitro culture - that it requires a number of Major Revisions.
My suggested changes and reviewer comments are shown below and on your article 'Overview' screen.
Please address these changes and resubmit. Although not a hard deadline please try to submit your revision within the next 35 days.


Editor comments:

Also, please address the following revisions;
- The English language of the whole manuscript needs editing and revision

- Important biochemical analyses are still needed, including estimation of the activity of primary metabolic enzymes (carbonic anhydrase, nitrate reductase, etc) to have a much clearer picture regarding the positive effects of Si on the performance of the plant in question.

Reviewer 1 ·

Basic reporting

See the attached file.

Experimental design

See the attached file

Validity of the findings

See the attached file

Additional comments

See the attached file

Annotated reviews are not available for download in order to protect the identity of reviewers who chose to remain anonymous.

·

Basic reporting

Line no. 75-76: The scientific name of the strawberry? please check

Experimental design

its written well

Validity of the findings

Line No. 107-109: if Si is already abundant in sand, why you want to apply exogenously and test against abiotic stresses

All the references are not as per journal style

Additional comments

what is the significance of choosing Aechmea blanchetiana as model plant?

If Aechmea blanchetiana is naturally growing in salinity rich sandbank area, it might have well adopted to the salinity. What is the logic in utilizing Si for salinity tolerant.

Reviewer 3 ·

Basic reporting

(a) In the introduction section there is no need to explain the concept of chlorophyll fluorescence.
(b) Discussion section is poorly written. The author did not back up his findings with suitable references.

Experimental design

(a) Critical information is missing from the methodology description of chlorophyll fluorescence parameters.
(b) Why a different number of replicates was used for different parameters? In some places, 5 replication number (line no. 210), and in some places 12 has been mentioned (line no. 203).
(c) In the material and method section (line no 126), it is mentioned that there was a total of eight treatments, but in line no. 215 and 216 authors mentioned that the total number of treatments was six.

Validity of the findings

(a) On what basis are doses of NaCl and Si finalized? Did the authors perform any screening experiments?

Additional comments

(a) The paper lacked some key biochemical index data of plants viz. the authors should have estimated the activity of primary metabolic enzymes (carbonic anhydrase, nitrate reductase, etc) to have a much clearer picture regarding the positive effects of Si on the performance of the plant in question.
(b) How can the authors justify that the effect was due to Si and not because of Ca as the authors applied Si in the form of CaSiO3?

---

## Round 0.2 · accepted · Accept

Authors made the required changes.

Reviewer 3 ·

Basic reporting

Changes accepted

Experimental design

Revision accepted

Validity of the findings

Valid findings

Additional comments

Authors made the required changes and explained the queries put up.